# Promoting Photosynthetic Production of Dammarenediol-II in *Chlamydomonas reinhardtii* via Gene Loading and Culture Optimization

**DOI:** 10.3390/ijms241311002

**Published:** 2023-07-02

**Authors:** Mei-Li Zhao, Xiang-Yu Li, Cheng-Xiang Lan, Zi-Ling Yuan, Jia-Lin Zhao, Ying Huang, Zhang-Li Hu, Bin Jia

**Affiliations:** 1Guangdong Technology Research Center for Marine Algal Bioengineering, Guangdong Provincial Key Laboratory for Plant Epigenetics, Shenzhen Engineering Laboratory for Marine Algal Biotechnology, Longhua Innovation Institute for Biotechnology, College of Life Sciences and Oceanography, Shenzhen University, Shenzhen 518060, China; zml2xly@163.com (M.-L.Z.); leerduo727@163.com (X.-Y.L.); lanchengxiang96@163.com (C.-X.L.); m18064628957@163.com (Z.-L.Y.); 13692181182@163.com (J.-L.Z.); huangy@szu.edu.cn (Y.H.); huzl@szu.edu.cn (Z.-L.H.); 2College of Physics and Optoelectronic Engineering, Shenzhen University, Shenzhen 518060, China; 3Bamboo Industry Institute, Zhejiang A&F University, Lin’an 311300, China

**Keywords:** ginsenosides, microalgae, *Chlamydomonas reinhardtii*, dammarenediol-II, metabolic engineering

## Abstract

Ginsenosides are major bioactive compounds found in *Panax ginseng* that exhibit various pharmaceutical properties. Dammarenediol-II, the nucleus of dammarane-type ginsenosides, is a promising candidate for pharmacologically active triterpenes. Dammarenediol-II synthase (DDS) cyclizes 2,3-oxidosqualene to produce dammarenediol-II. Based on the native terpenoids synthetic pathway, a dammarane-type ginsenosides synthetic pathway was established in *Chlamydomonas reinhardtii* by introducing *P. ginseng PgDDS*, CYP450 enzyme (*PgCYP716A47*), or/and *Arabidopsis thaliana* NADPH-cytochrome P450 reductase gene (*AtCPR*), which is responsible for producing dammarane-type ginsenosides. To enhance productivity, strategies such as “gene loading” and “culture optimizing” were employed. Multiple copies of transgene expression cassettes were introduced into the genome to increase the expression of the key rate-limiting enzyme gene, *PgDDS*, significantly improving the titer of dammarenediol-II to approximately 0.2 mg/L. Following the culture optimization in an opt2 medium supplemented with 1.5 mM methyl jasmonate under a light:dark regimen, the titer of dammarenediol-II increased more than 13-fold to approximately 2.6 mg/L. The *C. reinhardtii* strains engineered in this study constitute a good platform for the further production of ginsenosides in microalgae.

## 1. Introduction

Plant-derived natural terpenoids are considered important pharmaceuticals due to their anti-cancer properties (such as taxol and ginsenosides), anti-inflammatory effects (tanshinones), antimalarial activity (artemisinin), and antioxidative properties (including ginsenosides) [1,2,3,4,5]. However, there is a high cost of direct extraction from plants, coupled with low quantities of the target molecule, fluctuating yields, and negative environmental impact. Moreover, the complexity and chirality of natural terpenoids hinder their production through chemical synthesis [6]. For instance, ginseng roots require approximately 6 years of growth until harvest, experiencing varying influences related to weather, soil, and the presence of pathogens. However, the total ginsenosides contents in 5- to 7-year-old *Panax ginseng* roots are approximately 2% g/g dry weight, and some rare ginsenosides account for less than 0.01%, making it time-consuming and unsustainable by direct extraction. Moreover, due to the stereo-chemical complexity of ginsenosides, it is also challenging to synthesize by a chemical method. The advancement of synthetic biology has made isoprenoid engineering in microorganisms an alternative and attractive route to meet the increasing demand for natural terpenoids [6]. Traditionally, synthetic biology-based microbe cell factories, including bacteria and yeasts, were used to produce high-value plant natural terpenoids [7].

Ginsenosides, a triterpenoid glycoside, are primary bioactive components of ginseng roots [4,5,8,9]. Because their diverse pharmacological activities, such as anticancer, antioxidant, antiaggregant, antimicrobial, prevention of cardiovascular disease, improving immune function, and prevention of coronavirus disease 2019 (COVID-19), are making the ginsenosides commercially interesting [8]. Approximately 300 ginsenosides have been identified from the *Panax* species. According to the percentages of the total ginsenoside content, ginsenosides can be classified into major ginsenoside and minor ginsenoside groups. The minor ginsenosides show great biological and pharmacological activities, such as F1, F2, Rg2, Rg, Rh2, and CK, which have higher anticancer activities than the major ginsenosides. On the other hand, according to their chemical structure, ginsenosides can be divided into four groups: protopanoxadiol (PPD) group, protopanaxatriol (PPT), ocotillol group, and oleanane ginsenoside group. Moreover, according to their skeletons, ginsenosides can be divided into dammarane- and oleanane (OA)-type saponins. Dammarane-type ginsenosides include PPD-, protopanaxatriol (PPT)-, and ocotillol (OCT)-type [5,10,11,12].

The biosynthesis of ginsenosides involves several key steps: formation of ginsenoside skeletons, synthesis of sugar donors, and modification of the skeletons. The process begins with the sequential catalysis of two molecules of IPP and DMAPP to form FPP, followed by the catalysis of two FPP molecules to form squalene. Squalene is then oxygenated to form the key precursor of ginsenosides, (3S)-2,3-oxidosqualene. Dammarenediol-II is cyclized by dammarenediol-II synthase (DDS) and β-amyrin is catalyzed by β-amyrin synthase (β-AS) to biosynthesize dammarane-type and OA-type ginsenosides, respectively. Three characterized CYPs from the CYP716A subfamily have been identified in *Panax ginseng* and are responsible for diversifying ginsenoside structures. PPD is produced by the oxidation of dammarenediol-II by CYP716A47 (PPDS), and PPT is generated by the hydroxylation of PPD by CYP716A53v2 (PPTS). Oleanolic acid biosynthesis is catalyzed by CYP716A52v2 (OAS) (Figure 1A). The structural diversity of ginsenosides is further increased through the glycosylation of triterpene scaffolds by UDP-dependent glycosyltransferases (UGTs) using UDP-sugars. Finally, malonylation at the hydroxyl groups of sugar chains generates malonyl-ginsenosides [8,12].

Ginsenosides are traditionally extracted from ginseng roots. However, limitations such as the scarcity of wild ginseng roots, time-consuming cultivation, environmental influences on quantity and quality, and low growth rate and ginsenosides content in cell cultures have prompted the exploration of alternative approaches for mass production. One such approach is the construction of microbial cell factories [6,8]. Recently, eukaryotic microalgae have emerged as sustainable alternatives for biotechnological production processes [13,14,15]. Light-driven photosynthetic microbes, such as *Chlamydomonas reinhardtii* can offer the potential for sustainable production processes [16,17]. Algal cells are ideal hosts for the production of heterologous plant terpenoids owing to their shared distant evolutionary ancestry with land plants and their cultivation temperatures similar to those used for plant growth [18].

The construction of microalgae cell factories can provide an alternative approach to producing ginsenosides, but it has not been reported until now. In the present study, we established the biosynthetic pathways of dammarane-type ginsenosides in *C. reinhardtii* and subsequently applied multistep strategies to increase the titers of dammarenediol-II. These strategies included introducing *Arabidopsis thaliana* NADPH-cytochrome P450 reductase gene (CPR) to make more carbon flux flow to dammarane-type ginsenosides, gene-loading to introduce multiple copies of a transgene expression cassette into the genome, and optimizing the cultural and extraction condition (Figure 1B). The dammarenediol-II titers of strain SV3 reached approximately 0.4 and 1.3 mg/L when cultured with TAP and opt2 medium, respectively. After evaluating the culture parameters and treating methyl jasmonate (MeJA) (1 mM), the dammarenediol-II titer reached approximately 2.6 mg/L.

## 2. Results

### 2.1. Construction of Dammarane-Type Ginsenosides Synthetic Pathway in C. reinhardtii

In order to construct the dammarane-type ginsenosides synthetic pathway in *C. reinhardtii*, the codon-optimized *PgDDS,* and *PgCYP716A47* of *P. ginseng* [6] were integrated into the chromosome of *C. reinhardtii*. As Zhao, et al. [19] reported, the fusion of enzymes in a single polypeptide can express functional products, but the yield of the fusion enzyme is lower than that of the enzyme expressed individually in *C. reinhardtii*. Thus, in this study, we constructed the *P_Psad_-PgDDS-T_Psad_* and *P_FDX_-PgCYP716A47-T_FDX_* cassettes and then fused them in a V1 vector. The codon-optimized *PgDDS* and *PgCYP716A47* were controlled by *PSAD* and *FDX* promoters, respectively (Figure 2A). These two strong promoters are constitutive and compatible with *C. reinhardtii* fermentation [20,21]. After spectinomycin resistance selection, we obtained approximately 800 transformants. Then, colonies were picked from the plate and 124 transformants were verified by PCR amplification. In order to obtain high-expression strains, we further screened the transformants by qRT-PCR. The strains, SV1-12, SV1-25, and SV1-33 (transformants that contain *P_Psad_-PgDDS-T_Psad_* and *P_FDX_-PgCYP716A47-T_FDX_* cassettes), have higher expression levels of *PgDDS* and *PgCYP716A47* compared with other SV1 strains. However, the gene expression levels of *PgDDS* and *PgCYP716A47* were still considerably low when compared to the endogenous *actin* gene (Figure 2B). Terpene production was promoted using two-phase extractive fermentation. Dodecane was selected as the extracting solvent due to its low volatility, compatibility with algal growth, and ability to solvate terpenoids [22]. The dodecane phases were analyzed using gas chromatography–mass spectroscopy (GC-MS). The results revealed that the retention times of the standard dammarenediol-II and PPD peaks were 38.10 (Figure 3A) and 42.46 (Appendix A) min, respectively. The mass spectra of SV1 are the same as the authentic dammarenediol-II standard (Figure 3B,C). No PPD signal was detected among all of the detected strains (Figure 3C). The R^2^ coefficient for the standard calibration curves of dammarenediol-II was greater than 0.99 (standard curve in Appendix A). After GC-MS determination, the titer of dammarenediol-II was approximately 30 µg/L (Figure 2C).

In order to improve the titer of dammarenediol-II and PPD, the NADPH-cytochrome P450 reductase (*CPR*) gene, *A. thaliana* (*AtCPR*) was introduced. Dai, et al. [6] reported that the redox partner, a P450 enzyme, is very important for PPD synthesis. We constructed the *P_Psad_-PgDDS-T_Psad_*, *P_FDX_-PgCYP716A47-T_FDX_*, and *P_FDX_-AtCPR-T_FDX_* cassettes, and then fused them in the V2 vector (Figure 4A). After zeocin resistance selection, we obtained approximately 800 transformants. Then, colonies were picked from the plate and verified by PCR amplification, and 139 transformants were verified. In order to obtain high-expression strains, we further screened the transformants by qRT-PCR methods. The strains, SV2-9, SV2-17, and SV2-22 (transformants which contain *P_Psad_-PgDDS-T_Psad_*, *P_FDX_-PgCYP716A47-T_FDX_*, and *P_FDX_-AtCPR-T_FDX_* cassettes), have higher expression levels of *PgDDS*, *PgCYP716A47*, and *AtCPR* among all of the SV2 strains. The expression levels of *PgDDS* and *PgCYP716A47* in SV2 strains are also higher than those in SV1 strains (Figure 4B–D). After GC-MS determination, the titer of dammarenediol-II was approximately 70 µg/L of SV2 strains, approximately 2-fold higher than SV1 strains (Figure 4E). The PPD titer was not detected among all of the SV2 strains. As mentioned above, we concluded that the expression level of key genes of the dammarane-type ginsenosides biosynthesis pathway was insufficient to produce the higher titer dammarenediol-II and PPD in *C. reinhardtii*.

### 2.2. Dammarenediol-II Production Optimization

#### 2.2.1. Employed Gene Loading Strategy to Increase Dammarane-Type Ginsenosides Synthesis

In order to improve the expression level, we introduced multiple copies of transgene expression cassettes introduced into the genome. We co-expressed the V1 and V2 in *C. reinhardtii* by sequential transformation to achieve higher expression of the dammarane-type ginsenosides biosynthesis synthase (Figure 5A). After integrating these two vectors into the chromosome of *C. reinhardtii*, the TAP plates supplemented with antibiotics such as 10 mg L^−1^ zeocin and 200 mg L^−1^ spectinomycin were used to select the transformants. Then, approximately 1000 colonies were picked from the plate and verified by PCR amplification. In order to obtain high-expression strains, we further screened the transformants by qRT-PCR methods. As we expected, the expression levels of *PgDDS* and *PgCYP716A47* in some of the SV3 strains (SV3-15, 28, 40, 43, 135, and 141 transformants, which co-expressed V1 and V2), were approximately 10-fold higher than in the SV2 strain (Figure 5B,C). Furthermore, the proteins exhibited appropriate molecular masses, PgDDS (~90 kDa) and PgCYP716A47 (~56 kDa), in western blotting (Figure 5E). The SV3 strains, which contained the V1 and V2 constructs, produced the highest dammarenediol-II amount, achieving approximately 0.4 mg/L (Figure 6A). We have examined the PPD using the GC-MS method but did not detect any signal of the PPD.

#### 2.2.2. Culture Medium Optimization

Previous reports have proven that opt2 medium, which increased amounts of acetic acid, could both promote cell growth and increase the zeaxanthin and limonene production in *C. reinhardtii* [19,23]. Therefore, we cultured the higher dammarenediol-II-producing strains, SV3-43, SV3-135, and SV3-141, with the opt2 medium and TAP medium, to verify which medium is better for dammarenediol-II production. The results showed that the strains, SV3-43, SV3-135, and SV3-141 cultured with opt2 medium obtained titers of approximately 1.5, 1.5, and 1.3 mg/L, respectively, approximately 3.5-fold higher than the same strains were cultured with TAP medium (Figure 6A).

#### 2.2.3. Light Regime Optimization

Scale-up cultivations of the best dammarenediol-II-producing strain analyzed in Figure 6A were performed. Cultivations were conducted with either 16:8 h light:dark (L:D, LA) cycles or continuous light as previously described [19]. We carried out a scaled-up cultivation of the highest dammarenediol-II producing strains, SV3-43, SV3-135, and SV3-141, in an opt2 medium under both constant (24 h) light and 16:8 h L:D cycle conditions. After analyzing products, we found that the L:D cycle condition is better for dammarenediol-II production (Figure 6B). The dammarenediol-II titer of the SV3-43, SV3-135, and SV3-141 strains was approximately 1.5, 1.5, and 1.3 mg/L, respectively, when grown under L:D cycling conditions, approximately 7-fold higher than the same strains grown with continuous light condition.

#### 2.2.4. Extraction Solvent Optimization

Zhao, et al.’s [19] research showed that the length of the dodecane overlay treatment influences limonene production. Thus, we performed scaled-up cultivation in an opt2 medium of the three better dammarenediol-II-producing strains and collected the dodecane fractions after 4, 5, 6, 7, 8, 9, and 10 d growth (Figure 6C). We observed the highest dammarenediol-II accumulation after 7 d, with titers of up to 1.5, 1.5, and 1.3 mg/L from strains SV3-43, SV3-135, and SV3-141, respectively (Figure 6C).

Overmans and Lauersen [24] recently discovered that a synthetic perfluorocarbon liquid (FCs) can act as physical sinks for microbially produced isoprenoid compounds. FCs are stable and inert, and are promising alternatives to traditional solvents. To compare the efficiencies of extraction between dodecane and FCs, we cultured the strains, SV3-43, SV3-135, and SV3-141, with opt2 medium supplementary with dodecane or FCs as extract solvent. We did not detect the titer of dammarenediol-II and PPD when using FCs as extract solvent.

#### 2.2.5. Employed MeJA to Increase Dammarane-Type Ginsenosides Synthesis

Treatment with MeJA resulted in the increased intracellular amount of squalene, (S)-2,3-epoxysqualene, and cycloartenol in *C. reinhardtii* [25]. Squalene and (S)-2,3-epoxysqualene are precursors for the biosynthesis of dammarenediol-II and PPD. The strains were treated in the mid-exponential growth phase (cell density of ~5 × 10^6^ cells mL^−1^) with 1 mM, 1.5 mM, or 2 mM MeJA. After analyzing dammarenediol-II and PPD production, the dammarenediol-II titer of the SV3-43, SV3-135, and SV3-141 strains was approximately 1.8, 2.3, and 1.7 mg/L, respectively, when cultured with 1 mM MeJA; approximately 2.2, 2.8, and 3.3 mg/L, respectively, when cultured with 1.5 mM MeJA; and approximately 0.2, 0.4, and 0.3 mg/L, respectively, when cultured with 2 mM MeJA (Figure 6D). A PPD signal was not detected in any of the tested strains.

### 2.3. Metabolomic Profiling of Transgenic Algae

To further understand the differences in metabolite composition of the transgenic algae, we compared metabolomic changes (datasets obtained from UHPLC-MS/MS and GC-MS) between SV3 and UVM4. Based on the local metabolite database, qualitative and quantitative mass spectrometry analyses were conducted on the metabolites in the samples. In total, 120 terpenoids were identified in datasets obtained from UHPLC-MS/MS, including 12 monoterpenoids, 38 sesquiterpenoids, 29 diterpenoids, 30 triterpenes, and 11 terpenes. In addition, 409 metabolites were identified in datasets obtained from GC-MS, including 7 acids, 5 amines, 41 alcohols, 38 aldehydes, 22 aromatics, 1 ether, 60 esters, 4 halogenated hydrocarbons, and 64 heterocyclic compounds, 52 hydrocarbons, 31 ketones, 10 phenols, 7 sulfur compounds, 64 terpenoids, and 3 other metabolites. Detailed information on all identified metabolites is shown in Appendix A.

Based on fold change ≥ 2 or ≤0.5 and VIP ≥ 1, there were 19 and 46 significantly different metabolites, which were identified from UHPLC-MS/MS (4 upregulated, 15 down-regulated) and GC-MS (34 upregulated, 12 down-regulated), between SV3 (SV3-141) and UVM4, respectively (Figure 7 and Appendix A). In particular, some of the significantly different metabolites, which belong to terpenoids, have medicinal properties.

## 3. Discussion

In order to provide an alternative approach to producing ginsenosides, we established the biosynthetic pathways of dammarane-type ginsenosides in *C. reinhardtii* and subsequently applied multistep strategies to increase the titers of dammarenediol-II. *C. reinhardtii* are interesting candidates for the production of heterologous isoprenoid products because of the characteristics of controllability under laboratory conditions, fast growth with high cell density cultivation, well-characterized genetics, and well-developed genetic manipulation technology. The microalgal cellular environment is more favorable to the plant terpene synthases (TPSs) than bacterial, yeast, or cyanobacterial hosts because they share evolutionary ancestry with land plants [17].

As Zhao, et al. [19] reported, the fusion of enzymes in a single polypeptide can express functional products, but the yield of the fusion enzyme is lower than that of the enzyme expressed individually in *C. reinhardtii*. Hu, et al. [26] also reported that the fusion reduces the distance between the respective enzymes, and the folding and natural conformation of the multidomain protein can be impaired compared to the free enzymes, and thus can hinder their catalytic ability. In addition, Lauersen [17] reported that the large size of such constructs can also cause false-positive transformants in *C. reinhardtii*. Thus, in this study, we constructed the *P_Psad_-PgDDS-T_Psad_*, *P_FDX_-PgCYP716A47-T_FDX_*, and/or *P_FDX_-AtCPR-T_FDX_* cassettes, and then fused them in one vector to avoid such problems. In *C. reinhardtii*, transgene expression from the nuclear genome generally results in low amounts of recombinant protein [27]. Researchers always overexpress target proteins by generating multi-copy clones, which is gene loading [28]. Lauersen, et al. [29] used the gene loading strategy to increase total patchoulol yield, which co-overexpressed the PcPs-CFP in the PcPs-YFP #18 strain, by means of increasing the expression of the PcPs synthase. In this research, we also used the gene loading strategy and co-overexpressed V1 and V2 in *C. reinhardtii* to improve the expression of *PgDDS* and *PgCYP716A47* (Figure 5B,C). Consequently, this strategy successfully increased production of dammarenediol-II to 0.4 mg/L in the SV3 strain (transgenic algae harboring V1 (*P_Psad_-PgDDS-T_Psad_* and *P_FDX_-PgCYP716A47-T_FDX_* cassettes) and V2 (*P_Psad_-PgDDS-T_Psad_*, *P_FDX_-PgCYP716A47-T_FDX_*, and *P_FDX_-AtCPR-T_FDX_* cassettes) constructs) (Figure 6A).

The *C. reinhardtii* can grow by using light for photoautotrophic growth, by using acetate as a sole carbon source for heterotrophic growth, and by using light and acetate for mixotrophic growth. Acetate has an important effect on the biomass and heterologous protein production of *C. reinhardtii*. Song, et al. [23] showed that it will promote cell growth and increase zeaxanthin production in *C. reinhardtii* by using an opt2 medium. Zhao, et al.’s [19] research has verified that limonene production can be increased by culturing *C. reinhardtii* with an opt2 medium. As shown in Figure 6A, the production of dammarenediol-II increased 3.5-fold by using the opt2 medium compared with the TAP medium. Except for acetate, the light regime has a great impact on productivity, when *C. reinhardtii* is under mixotrophic growth. Previous research showed that L:D cycles prolong the exponential growth phase, increasing the productivity of *C. reinhardtii* [19,29,30]. As expected, the production of dammarenediol-II is also higher in L:D cycles than in the constant light regime (Figure 6B). Commault, et al. [25] discovered new extract solvents, FCs, that can be used as bio-compatible liquids for microbial cell milking of heterologous isoprenoids. FCs are the same as traditional solvents, such as dodecane; they are all stable and inert. Moreover, the research demonstrated that the FCs provide cleaner extraction of isoprenoid products from microbial culture than dodecane. However, most FCs were less efficient in isoprenoid extraction from the algal cells than dodecane except for the capture of diterpenoids manoyl oxide by FC-770. We found that FC was less efficient in dammarenediol-II extraction. The length of the dodecane overlay treatment is essential for the best product titers [31]. When capturing the sesquiterpenoid patchoulol by dodecane, Lauersen, et al. [29] observed that the patchoulol accumulation increased in the dodecane fraction even after the cells had reached the stationary phase. We collected the dodecane fractions after 4, 5, 6, 7, 8, 9, and 10 d growth; the accumulation of dammarenediol-II is highest extracted by dodecane when harvested at 7 d (Figure 6C).

As previously reported, the addition of exogenous MeJA can upregulate key genes of the 2-C-methyl-D-erythritol 4-phosphate (MEP) pathway, leading to a significant increase in intermediates of this pathway, squalene, and (S)-2,3-epoxysqualene [22,25]. As mentioned in Jia, et al. [32], after exogenous application of MeJA for 24 h, the key genes of the MEP pathway (DXS, DXR, and CMK), FPPS, and SQE are significantly increased. These data indicate the redirection of the carbon flux toward the synthesis precursors of triterpenoid secondary metabolites upon MeJA treatment. In *P. ginseng*, MeJA treatment can transcriptionally activate the *CYP716A47* gene and enhance ginsenoside biosynthesis. As shown in Liu, et al. [33], the expression level of both *PgMYB2* and *PgDDS* significantly increased to the highest point after 24 h treatment of MeJA. These data indicate the MeJA treatment can redirect carbon flux toward the synthesis of dammarane-type ginsenosides. In order to increase productivity, we added the MeJA to enhance the amount of squalene and (S)-2,3-epoxysqualene [22,25,32]. The production of dammarenediol-II increased 7-fold after 1.5 mM MeJA treatment (Figure 6D). Although, the exogenous application of MeJA can increase the amount of squalene and (S)-2,3-epoxysqualene [22,25], the accumulation of PPD is also below detection levels. As displayed in Figure 5E, each construct expressed successfully, and signals were observed to exhibit appropriate molecular mass from the PgDDS enzyme in total cellular protein samples by western blotting, and SV3-141 has successfully expressed appropriate molecular mass from the PgCYP716A47 enzyme in total cellular protein samples by western blotting. However, we did not detect the signal of PPD. We presumed that the large size of constructs (V1 and V2) may affect the folding and natural conformation of the multidomain protein. Dai, et al. [6] also reported that engineered yeast strains can accumulate large amounts of dammarenediol-II, but could not increase protopanaxadiol production by increasing copy numbers of *PPDS* (also known as *CYP716A47*) and *CPR*. In addition, the heterologous expression of *CYP716A47* in microorganisms results in low coupling between *CPR*, leading to reduced growth and terpenoid synthesis [8]. A future study should modify the transmembrane domain truncation and self-sufficient CYP716A47-CPR fusion construction to increase the protopanaxadiol production.

At present, the engineered microorganisms, which are used for ginsenoside biosynthesis, include bacteria and yeasts. Bacteria produce ginsenoside mainly via recombinant β-glucosidase or uridine diphosphate glycosyltransferase (UGTs) [8]. The most general strategies for yeasts to produce ginsenoside are through heterologous gene expression and enzyme engineering [9]. A cell suspension culture of transgenic tobacco can produce 5.2 mg/L dammarenediol-II [34]. Transgenic rice can produce 0.44 mg/g dw dammarenediol-II, 0.59 mg/g dw PPD, and 0.43 mg/g dw protopanaxatriol (PPT) [35]. The volumetric production of dammarenediol-II by feeding with squalene was up to 13.233 mg/L under the common shake flask fermentation conditions of transgenic *Pichia pastoris* [36]. The highest dammarenediol-II titer was approximately 3.3 mg/L in transgenic algae when cultured with 1.5 mM MeJA in this study (Table 1), which indicates the *C. reinhardtii* strains engineered in this study constitute a good platform for further production of ginsenosides.

## 4. Materials and Methods

### 4.1. C. reinhardtii Strain and Cultivation Conditions

We used the *C. reinhardtii* UV-mutated 4 (UVM4, cell wall-deficient) strain for all of the experiments in this work because of its capability of efficiently expressing transgenes and overcoming the long-standing obstacle of the disappointingly poor expression of transgenes in the algal nuclear genome [37]. The UVM4 algal species that originated from Prof. Dr. Ralph Bock can grow well in freshwater environments. Unless otherwise noted, we maintained this algal strain in continuous light (150 µmol m^−2^ s ^−1^) at a 25 °C, environment and cultured this strain by using a tris-acetate-phosphate (TAP) medium [38] on agar plates or liquid culture. The transformants were maintained in the TAP plates supplemented with antibiotics such as 10 mg L^−1^ zeocin. For the liquid cultures, microtiter plates or erlenmeyer flasks containing TAP without antibiotics were used.

### 4.2. Plasmid Construction

Thermo Fisher Scientific FastDigest restriction enzymes (Thermo Fisher Scientific, Shanghai, China), pEASY^®^-Blunt Zero Cloning Kit (Transgen Biotech, Beijing, China), and ClonExpress^®^ II One Step Cloning Kit (Vazyme, Nanjing, China) were used in this work for vector construction, according to the manufacturer’s protocols. Following the manufacturer’s protocols, KOD One^TM^ PCR Master Mix -Blue- (TOYOBO, Shanghai, China) was used for performing all PCRs. The primers are listed in Appendix A. We confirmed the vector sequences by sequencing (Sangon Biotech, Shanghai, China) after each cloning step. *Escherichia coli* Top10 competent cells were used for plasmids transforming, and luria broth (LB) agar plates or liquid medium with 100 mg L^−1^ ampicillin were used for cultivating the transformants.

Amino acid sequences for the dammarenediol synthase from *P. ginseng* (PgDDS, ACZ71036) [6]; PPD synthase, which is a CYP enzyme, from *P. ginseng* (PgCYP716A47, AEY75212) [7]; and P450 reductase 2 from *A. thaliana* (AtCPR, NP_194750) [6] were codon optimized and synthesized de novo taking account of the nuclear codon bias of *C. reinhardtii* (General Biosystem Company, Anhui, China) and upload to NCBI (*PgDDS*: GenBank accession no. OQ980513, *PgCYP716A47*: GenBank accession no. OQ980514, and *AtCPR*: GenBank accession no. OQ980515). The expression boxes of the sequences mentioned above were fused together, then designed with the compatible restriction endonuclease sites, and cloned into pOpt expression vectors [39]. The PgDDS and PgCYP716A47 sequences were modified to contain an 8-amino-acid strep tag II peptide. All constructs and strains created are listed in Table 2 and Table 3, respectively.

### 4.3. Strain Construction

The glass bead agitation method was used to introduce the plasmids V1 or/and V2 into *C. reinhardtii* [40]. The TAP agar plates supplemented with 200 mg L^−1^ spectinomycin and/or 10 mg L^−1^ zeocin were used to select the positive transformants under a light intensity of 150 µmol photons m^−2^ s^−^1. We further confirmed the transformants using genomic PCR and qRT-PCR methods. The expression of the PgDDS and PgCYP716A47 reporter fusions to full length was verified via SDS PAGE and western blotting using α-StrepII tag-HRP linked antibody (IBA Lifesciences, Göttingen, Germany).

### 4.4. Two-Phase Extractive Fermentation of Positive Clone Strain

The assessment of PPD and its intermediate product, dammarenediol-II titer, was conducted for V1 genetic construct with three representative strains, each in biological triplicate. Seed culture was prepared by inoculating several colonies into a 500 mL flask containing 300 mL culture medium, and incubating at 25 °C and 120 rpm for 48 h until logarithmic phase. Then, the 5 mL dodecane overlay was added into the medium to perform two-phase extractive fermentation at 120 rpm, 25 °C, with 150 µmol photons m^−2^ s^−1^ 16:8 h light: dark (L:D) cycle conditions for 7 d. The dodecane fractions were collected by centrifugation at 8000 rcf for 6 min and then transferred to new sample tubes. After the filtration process, the quantification of PPD and its intermediate product, dammarenediol-II, in dodecane was conducted using GC-MS (Hewlett-Packard model 7890 A, Agilent Technologies, Santa Clara, CA, USA), equipped with a Rxi-5MS column (30 m × 0.25 mm × 0.25 µm, Restek, Alexandria, LA, USA). The detection method was performed as previously described [41]. The carrier gas was helium, and the injection volume was set to 1 µL (150 °C, 5 min; 5 °C/min to 300 °C, and hold 300 °C for 20 min). The commercial standard dammarenediol-II (YUANYE, Shanghai, China) and PPD (GLPBIO, Montclair, CA, USA) were used to confirm the retention time [41]. The quantification of limonene was performed using a standard curve of the commercial dammarenediol-II and PPD. The standard calibration curve was performed by using the range of 0.5–5 ppm dammarenediol-II and PPD in dodecane, and the R^2^ coefficient for the calibration was greater than 0.99 (standard curve in Appendix A). The mass spectrometer was set to full-scan and SIM mode, and the extracted-ion chromatograms (XIC) with mass ranges of 69, 109, and 207 for the dammarenediol-II determination and 69, 109, 135, 191, and 207 for the PPD determination. All measurements were performed in triplicate and the chromatograms were reviewed manually.

### 4.5. Optimization of the Fermentation and Dammarenediol-II Extraction Conditions

We optimized the fermentation and dammarenediol-II extraction conditions in order to enhance the dammarenediol-II production of the three highest-producing dammarenediol-II strains. To investigate the influence of light regimen, we culture the strains under constant (24 h) light or 16:8 h L:D cycle conditions. To investigate the influence of the medium, we culture the strains in a TAP medium or opt2 medium (TAP medium + 1 mL L^–1^ glacial acetic acid). To investigate the influence of solvent overlay, we culture the strains in an opt2 medium supplemented with dodecane or perfluoro-2-butyl tetrahydrofuran. To investigate the influence of the different dodecane extraction times (4, 5, 6, 7, 8, 9, and 10 d) on the dammarenediol-II production, the fermentation processes were performed in triplicate. At the end of the fermentation, the dodecane fractions were collected and assayed.

### 4.6. Statistical Analysis

The results are presented as the mean + standard deviation (SD). One-way ANOVA was used to analyze the significance of the differences, and the statistical significance was indicated by *p* < 0.05 or *p* < 0.01.

## Figures and Tables

**Figure 1 ijms-24-11002-f001:**
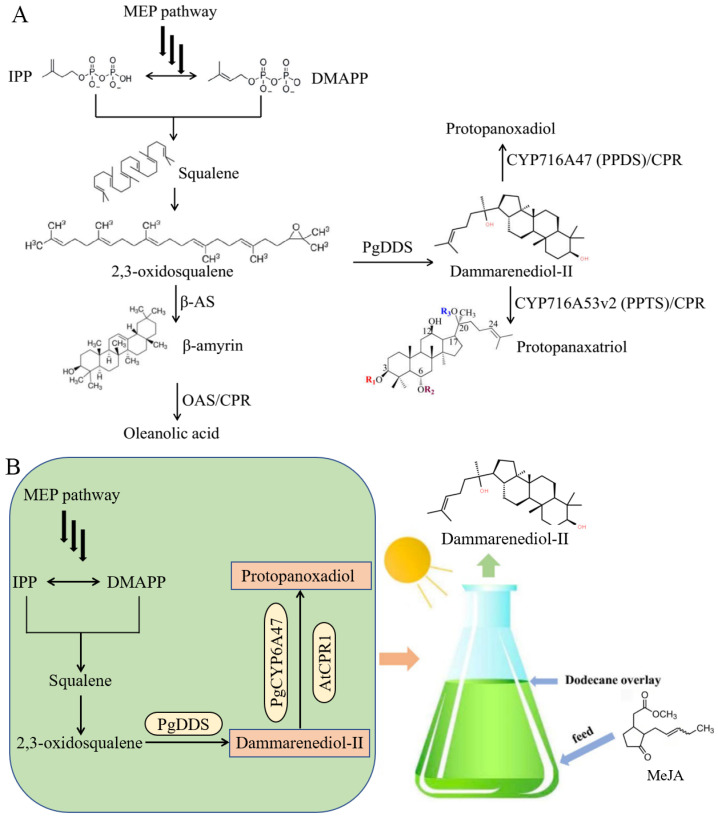
Biosynthesis pathways for ginsenoside production (**A**) and metabolic pathway design for the biosynthesis of dammarenediol-II and protopanoxadiol in engineered *Chlamydomonas reinhardtii* (**B**). AtCPR1: *Arabidopsis thaliana* NADPH-cytochrome P450 reductase 1; β-AS: β-amyrin synthase; CPR: cytochrome P450 reductase; DMAPP: dimethylallyl diphosphate; IPP: isopentenyl pyrophosphate; MEP: 2-C-methyl-D-erythritol 4-phosphate; OAS: oleanolic acid synthase; PgDDS: *Panax ginseng* dammarenediol synthase; PgCYPCYP6A47: *Panax ginseng* Cyt P450 (CYP) enzyme.

**Figure 2 ijms-24-11002-f002:**
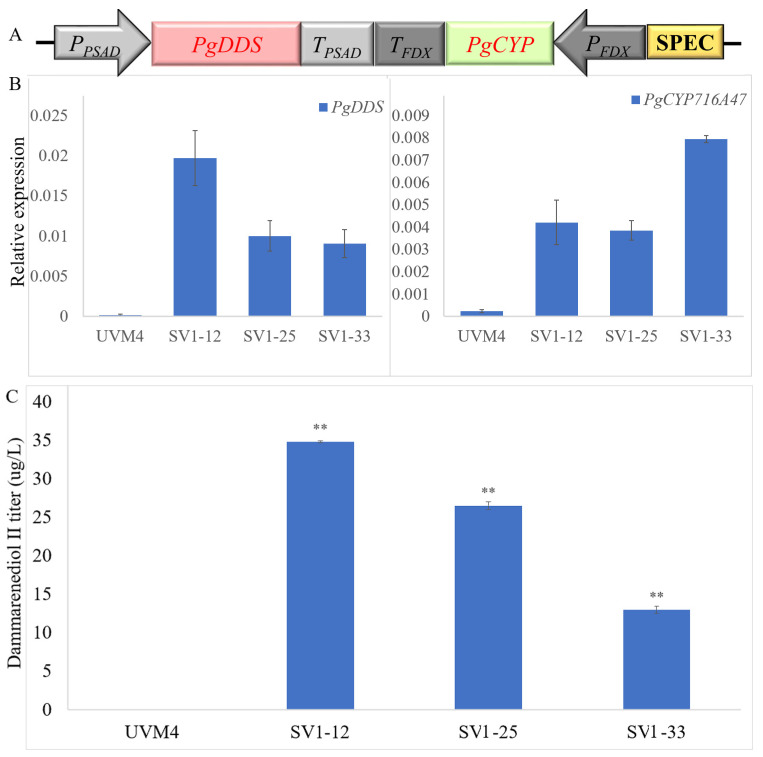
Overview of the expression and screening strategy for the transgenic *C. reinhardtii*. (**A**) Diagram of the codon-optimized *PgDDS* and *PgCYP716A47* gene expression cassettes; (**B**) Real-time fluorescence quantitative PCR analysis of the expression of *PgDDS* and *PgCYP716A47* in transgenic *C. reinhardtii* strains SV1. The qRT-PCR results were obtained from three independent biological replicates per sample. The levels of detected amplification were normalized via the amplified products of the *actin* genes as a reference. The error bars represent the standard errors. Statistical significance (one-way ANOVA) compared to UVM4 is represented by asterisks (** indicates a difference at the *p* ≤ 0.01 level); (**C**) Dammarenediol-II yield in transgenic algae.

**Figure 3 ijms-24-11002-f003:**
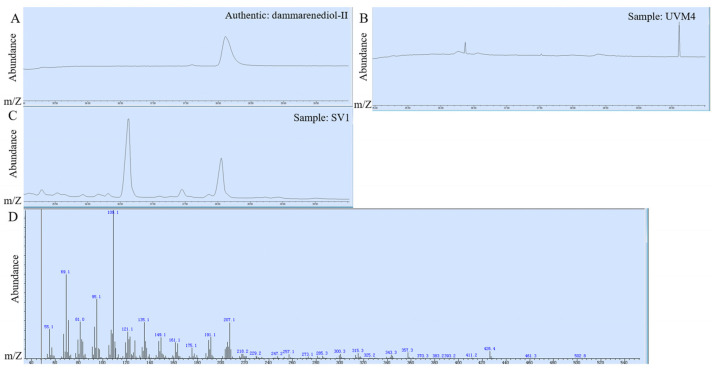
Identification of fermentation products of strain SV1-12. This strain was cultivated in a TAP medium. (**A**) GC-MS analysis of dammarenediol-II standard; (**B**) GC-MS analysis of cell extraction of the parent strain, UVM4; (**C**) GC-MS analysis of cell extraction of strain SV1; (**D**) mass spectra of dammarenediol-II.

**Figure 4 ijms-24-11002-f004:**
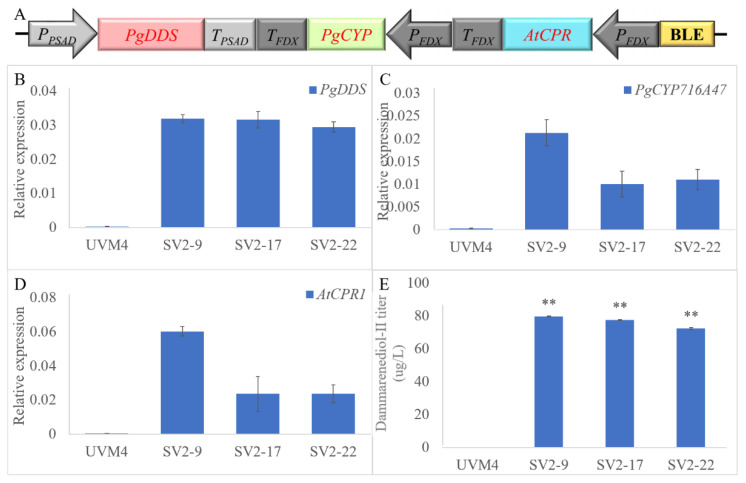
Overview of the expression and screening strategy for the transgenic *C. reinhardtii*. (**A**) Diagram of the codon-optimized *PgDDS*, *PgCYP716A47*, and *AtCPR* gene expression cassettes. BLE: bleomycin. Real-time fluorescence quantitative PCR analysis of the expression of *PgDDS* (**B**), *PgCYP716A47* (**C**), and *AtCPR* (**D**) in transgenic *C. reinhardtii* strains SV2. The qRT-PCR results were obtained from three independent biological replicates per sample. The levels of detected amplification were normalized via the amplified products of the *actin* genes as a reference. The error bars represent the standard errors. Statistical significance (one-way ANOVA) compared to UVM4 is represented by asterisks (** indicates a difference at the *p* ≤ 0.01 level). (**E**) Dammarenediol-II yield in transgenic algae.

**Figure 5 ijms-24-11002-f005:**
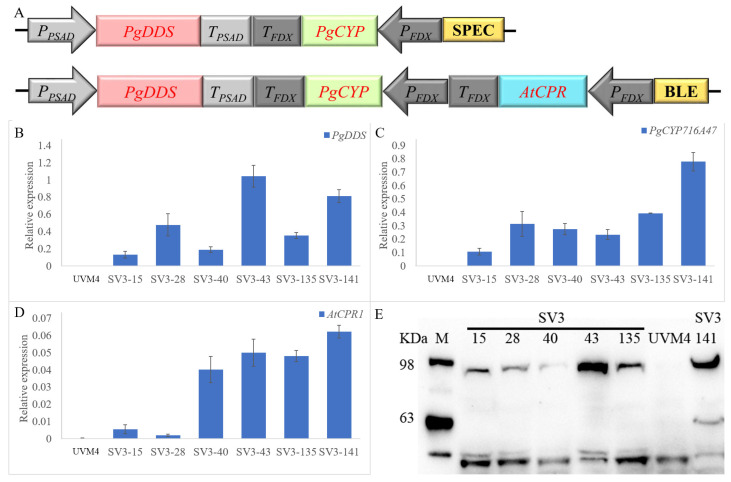
Overview of the expression and screening strategy for the transgenic *C. reinhardtii*. (**A**) Co-expression of V1 and V2 in *C. reinhardtii*. Real-time fluorescence quantitative PCR analysis of the expression of *PgDDS* (**B**), *PgCYP716A47* (**C**), and *AtCPR* (**D**) in transgenic *C. reinhardtii* strains SV3. The qRT-PCR results were obtained from three independent biological replicates per sample. The levels of detected amplification were normalized via the amplified products of the *actin* genes as a reference. The error bars represent the standard errors. (**E**) Western blot of a total cellular protein with an α-StrepII tag antibody. SV3-43, SV3-135, and SV3-141 expressing strains exhibit signals at the appropriate predicted molecular mass. Full-length expression of each protein (PgDDS (~90 kDa) and PgCYP716A47 (~56 kDa)) could be detected. M: marker.

**Figure 6 ijms-24-11002-f006:**
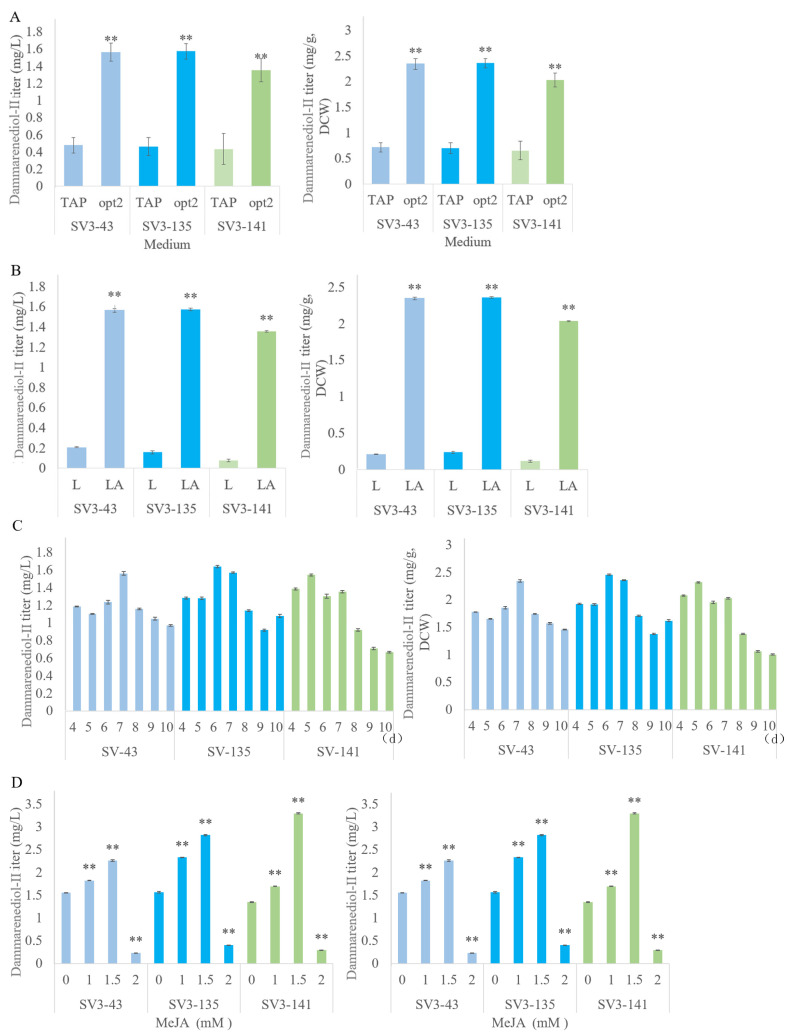
Dammarenediol-II yield in transgenic algae. (**A**) Strains were cultivated in 500 mL shake flasks containing a 300 mL TAP or opt2 medium at 25 °C and 120 rpm under 16:8 h light: dark (L:D) cycles (LA) condition, and using 5 mL dodecane as a solvent overlay for 7 d. (**B**) Strains were grown under constant (24 h) light (L) or 16:8 h L:D cycles (LA) conditions. Strains were cultivated in 500 mL shake flasks containing a 300 mL TAP medium at 25 °C and 120 rpm under constant (24 h) light or 16:8 h L:D cycles, and using 5 mL dodecane as an overlay for 7 d. (**C**) Strains were cultivated in 500 mL shake flasks containing a 300 mL TAP medium at 25 °C and 120 rpm under 16:8 h L:D conditions using 5 mL dodecane as an extraction overlay for 4, 5, 6, 7, 8, 9, or 10 d. (**D**) Strains were cultivated in 500 mL shake flasks containing a 300 mL TAP medium supplemented with 1, 1.5, or 2 mM MeJA at 25 °C and 120 rpm under 16:8 h L:D conditions, and using 5 mL dodecane as an extraction solvent for 7 d. Three biological replicates were used to calculate the dammarenediol-II yield averages and SDs. Statistical significance (one-way ANOVA) is represented by asterisks (** indicates a difference at the *p* ≤ 0.01 level).

**Figure 7 ijms-24-11002-f007:**
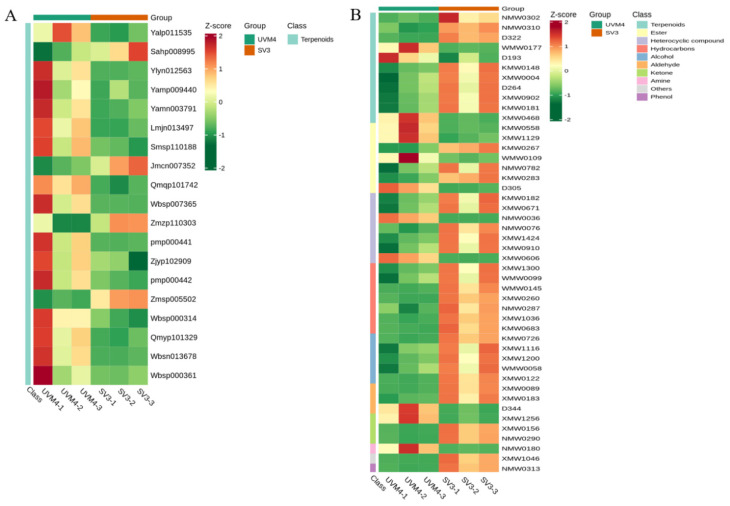
Heatmaps of the metabolites in transgenic algae obtained from UHPLC-MS/MS (**A**) and GC-MS (**B**).

**Table 1 ijms-24-11002-t001:** Summary of genetic engineering to yield dammarenediol-II in microorganisms.

Strain	Dammarenediol-II Titer	Cultivation Setup	Reference
Transgenic tobacco	5.2 mg/L	Cell suspension culture	[34]
Transgenic rice	0.44 mg/g	-	[35]
Transgenic *Pichia pastoris*	13.233 mg/L	Shake flask fermentation	[36]
Transgenic *C. reinhardtii*	3.3 mg/L	Shake flask fermentation with 1.5 mM MeJA	This study

**Table 2 ijms-24-11002-t002:** Genetic constructs used in this study.

Name	Description	Source	Antibiotic Resistance
pUC57	Cloning vector with multiple cloning sites	GenScript	Ampicillin
*PgDDS*	Cloning *PgDDS* gene into pUC57	This study	Ampicillin
*PgCYP716A47*	Cloning *PgCYP716A47* gene into pUC57	This study	Ampicillin
*AtCPR*	Cloning *AtCPR* gene into pUC57	This study	Ampicillin
pEASY-Blunt	Cloning vector with multiple cloning sites	TransGen Biotech	Ampicillin
p-*PgDDS*	Cloning *P_Psad_-PgDDS-T_Psad_* cassette into pEASY-Blunt	This study	Ampicillin
p-*PgCYP716A47*	Cloning *P_FDX_-PgCYP716A47*-T_FDX_ cassette into pEASY-Blunt	This study	Ampicillin
p-*AtCPR*	Cloning *P_FDX_-AtCPR-T_FDX_* cassette into pEASY-Blunt	This study	Ampicillin
pOpt	Cloning vector with multiple cloning sites	Lauersen, et al. [39]	Zeocin
pOpt-Ble	Cloning vector with multiple cloning sites	This study	Zeocin
V1	Cloning *P_Psad_-PgDDS-T_Psad_* and *P_FDX_-PgCYP716A47-T_FDX_* cassettes into pOpt2-Spec	This study	Spectinomycin
V2	Cloning *P_Psad_-PgDDS-T_Psad_*, *P_FDX_-PgCYP716A47-T_FDX_*, and *P_FDX_-AtCPR-T_FDX_* cassettes into pOpt2-Ble	This study	Zeocin

**Table 3 ijms-24-11002-t003:** Strains used in this study.

Name	Description	Source
SV1	Transformants that harbor V1 construct	This study
SV2	Transformants that harbor V2 construct	This study
SV3	Transformants that harbor V1 and V2 constructs	This study

## Data Availability

The data used to support the findings of this study are available from the corresponding author upon request.

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
