# Peer review of "Promoting Photosynthetic Production of Dammarenediol-II in Chlamydomonas reinhardtii via Gene Loading and Culture Optimization"

_ijms, 2023, doi:10.3390/ijms241311002_

Round 1

Reviewer 1 Report

The manuscript entitled “Promoting photosynthetic production of dammarenediol-II in Chlamydomonas reinhardtii via gene loading and culture optimization” describes possible production of ginsenoside using microalgae via gene loading approach and culture optimization. That can be a good approach to extend the strain development of industrial microalgae species. The manuscript can be further improved with some revision corresponding to the following comments:

1.      The reason of the use of spectinomycin as a selection force?

2.      Why did the author use UV-induced mutant of C. reinhardtii, not use a wild-type?

3.      A part of the sentence in line #143-144 is bold-marked.

4.      The author should check the Fig-numbering “Figure 5A” in line #209-210.

5.      For more clarity, it would be better to add “LA” abbreviation to the “L:D cycle condition” (as described in Figure 6 caption) in the sentences of Section 2.2.3.

6.      The author should revise the cell density concentration and the unit with upper case, in Line #237.

7.      Considering the paragraph in Line #358-369, presenting the comparison on ginsenoside biosynthesis from the engineered microorganisms as a table would make the manuscript content more effective.

8.      Spacing in Line #447

Author Response

manuscript ID: ijms-2449083

Metabolic Promoting Photosynthetic Production of Dammarenediol-II in Chlamydomonas reinhardtii via Gene Loading and Culture Optimization

Reviewers' comments:

Reviewer 1

Comments and Suggestions for Authors

The manuscript entitled “Promoting photosynthetic production of dammarenediol-II in Chlamydomonas reinhardtii via gene loading and culture optimization” describes possible production of ginsenoside using microalgae via gene loading approach and culture optimization. That can be a good approach to extend the strain development of industrial microalgae species. The manuscript can be further improved with some revision corresponding to the following comments:

  1. The reason of the use of spectinomycin as a selection force?

Response: The V1 vector contains the resistance gene of spectinomycin, so the strains,  transformed by this vector can be screened with spectinomycin.

  1. Why did the author use UV-induced mutant of C. reinhardtii, not use a wild-type?

Response: Because of unsuitable codon usage in the open reading frame (ORF), the nuclear expression levels of nondomestic cDNA genes are disappointingly poor in wild-type C. reinhardtii. The C. reinhardtii UV-mutated 4 (UVM4, cell wall-deficient) strain can efficiently express transgenes and overcome the long-standing obstacle of the disappointing poor expression of transgenes in the algal nuclear genome.

  1. A part of the sentence in line #143-144 is bold-marked.

Response: Thank you for your valuable suggestions. We have revised it.

  1. The author should check the Fig-numbering “Figure 5A” in line #209-210.

Response: Thank you for your valuable suggestions. We have revised it as “Figure 6A”.

  1. For more clarity, it would be better to add “LA” abbreviation to the “L:D cycle condition” (as described in Figure 6 caption) in the sentences of Section 2.2.3.

Response: Thank you for your valuable suggestions. We have added “LA” abbreviation.

  1. The author should revise the cell density concentration and the unit with upper case, in Line #237.

Response: Thank you for pointing out our neglect. We have revised it.

  1. Considering the paragraph in Line #358-369, presenting the comparison on ginsenoside biosynthesis from the engineered microorganisms as a table would make the manuscript content more effective.

Response: Thank you for your valuable suggestions. We have added the information in new Table 1.

  1. Spacing in Line #447

Response: Thank you for pointing out our neglect. We have revised it.

Reviewer 2 Report

Title: Promoting Photosynthetic Production of Dammarenediol-II in Chlamydomonas reinhardtii via Gene Loading and Culture Optimization

The paper describes a study that focuses on the production of dammarenediol-II, a triterpene compound found in Panax ginseng, using the microalgae Chlamydomonas reinhardtii as a production platform. The researchers introduced genes from Panax ginseng and Arabidopsis thaliana responsible for the synthesis of dammarane-type ginsenosides into Chlamydomonas reinhardtii to create a synthetic pathway for the production of these compounds. They employed strategies such as gene loading and culture optimization to enhance productivity.

The results showed that introducing multiple copies of the key enzyme gene, PgDDS, significantly increased the production of dammarenediol-II to approximately 0.2 mg/L. Furthermore, by optimizing the culture conditions using an opt2 medium supplemented with methyl jasmonate and a light:dark regimen, the titer of dammarenediol-II increased more than 13-fold to approximately 2.6 mg/L.

Overall, this study successfully engineered Chlamydomonas reinhardtii to produce dammarenediol-II, a promising candidate for pharmacologically active triterpenes found in Panax ginseng. The findings demonstrate the potential of microalgae as a platform for the production of ginsenosides. Further research and optimization could lead to the development of efficient and sustainable methods for producing these valuable bioactive compounds.

One potential improvement would be to provide more context regarding the significance of dammarenediol-II and dammarane-type ginsenosides in the pharmaceutical field. Additionally, it would be helpful to include any limitations or challenges encountered during the study. Overall, this paper provides a clear overview of the study's objectives, methods, and results, paving the way for further exploration in the field of ginsenoside production.

The review suggests a few improvements to enhance the paper's readability and flow before publication in the journal.
Introduction:

The introduction provides an overview of the importance of dammarenediol-II and ginsenosides as valuable compounds with various pharmaceutical and industrial applications. It mentions the challenges associated with traditional production methods and highlights the potential of microbial systems for efficient and sustainable production. The authors specifically focus on the microalga C. reinhardtii as a promising host organism due to its controllability, genetic characteristics, and well-established genetic manipulation techniques. The objectives of the study are clearly stated, emphasizing the use of gene loading and culture optimization to enhance dammarenediol-II production.

The introduction contains a few grammatical errors and areas that could be improved for clarity. Here are some suggested revisions:

·       "Plant natural terpenoids are seemingly important pharmaceuticals due to their anti-cancer (taxol and ginsenosides), anti-inflammatory (tanshinones), antimalarial (artemis- inin), and antioxidation (ginsenosides) properties [1–5]."

Revised: "Plant-derived natural terpenoids are considered important pharmaceuticals due to their anti-cancer properties (such as taxol and ginsenosides), anti-inflammatory effects (tanshinones), antimalarial activity (artemisinin), and antioxidative properties (including ginsenosides) [1–5]."

·       "However, the high price of direct extraction from plants due to low amounts of the target molecule, is fluctuating in both quantity and quality, and negative environmental impact."

Revised: "However, the high cost of direct extraction from plants, coupled with the low quantities of the target molecule, fluctuating yields, and negative environmental impact..."

·       "And the complexity and chirality of natural terpenoids hampered their production by chemical synthesis [6]."

Revised: "Moreover, the complexity and chirality of natural terpenoids hinder their production through chemical synthesis [6]."

·       "With the advance of synthetic biology, isoprenoid engineering in microorganisms is becoming an alternative and attractive route for addressing the increasing demand for natural terpenoids [6]."

Revised: "The advancement of synthetic biology has made isoprenoid engineering in microorganisms an alternative and attractive route to meet the increasing demand for natural terpenoids [6]."

1.       Could you provide more insight into the specific challenges and limitations encountered in the traditional extraction methods of ginsenosides that prompted the exploration of alternative approaches?

2.       How do the achieved dammarenediol-II titers in strain SV3 compare to previous studies or existing methods of ginsenoside production?

3.       Beyond ginsenosides, do you believe Chlamydomonas reinhardtii or similar microalgae platforms have potential for the production of other bioactive compounds, and if so, which ones and why?

Materials and Methods:

The materials and methods section describes the experimental procedures and techniques used in the study. It includes information on the construction of gene cassettes and vectors, transformation of C. reinhardtii, gene loading strategy, culture conditions, and extraction methods for dammarenediol-II. The section provides sufficient detail to allow for replication of the experiments.

Some sentences are overly complex or contain multiple ideas, making them difficult to follow. Try breaking them into shorter, more concise sentences to improve readability.

Here is a list of grammatical problems in the text:

In the sentence "because of its capable of efficiently expressing transgenes," the word "capable" should be "capability."

In the sentence "UVM4 algal species, which originated from Prof. Dr. Ralph Bock, can grow well in freshwater environments," the word "which" should be "that."

In the sentence "We maintained this algal strain in continuous light (150 µmol m−2 s −1 ) at 25℃ environment," there should be a comma after "25℃" and before "environment."

In the sentence "The amino acid sequences for the dammarenediol synthase from P. ginseng (PgDDS, ACZ71036) [6], the PPD synthase, which is a CYP enzyme, from P. ginseng (PgCYP716A47, AEY75212) [7], and the P450 reductase 2 form A. thaliana (AtCPR, NP_194750) [6] were codon optimized," "form" should be "from."

In the sentence "We confirmed the vector sequences by sequencing (Sangon Biotech, Shanghai, China) following each cloning step," "following" should be "after" to indicate the sequence of events.

In the sentence "The TAP agar plates supplemented with 200 mg L−1 spectinomycin or/and 10 mg L−1 zeocin were used to select the positive transformants under a light intensity of 150 µmol photons m−2 s−1," "or/and" should be "and/or."

In the sentence "We further confirmed the transformants by using the genomic PCR and qRT-PCR methods," "by using" can be simplified to "using."

In the sentence "The dodecane fractions were collected by centrifugation at 8000 rcf for 6 min, then transferred to new sample tubes," "then transferred" should be "and then transferred."

In the sentence "After filtrating, the quantification of PPD and its intermediate products dammarenediol-

In the sentence "The mass spectrometer was set to full-scan and sim mode," "sim mode" should be "SIM mode" or "selected ion monitoring mode."

In the sentence "All the fermentation processes were performed in triplicate," "All" can be removed to improve clarity.

Here are some questions related to this part of the Materials and Methods:

1.       Why was the C. reinhardtii UV-mutated 4 (UVM4) strain chosen for the experiments?

2.       Where did the UVM4 algal species originate from, and what are its characteristics in terms of growth and environmental requirements?

3.       What medium was used to culture the UVM4 strain, and what were the conditions for maintaining it?

4.       What were the specific modifications made to the PgDDS and PgCYP716A47 sequences, and why were they modified?

5.       What was the purpose of the two-phase extractive fermentation, and how was it performed?

6.       What optimization experiments were conducted to enhance dammarenediol-II production, and what were the specific conditions investigated?

It would be beneficial to mention the number of replicates performed for each experiment or assay. Additionally, providing information about the use of positive and negative controls would strengthen the experimental design.

7.       Were there any specific controls or replicates used in the experiments, and if so, what were they?

While the statistical analysis is mentioned, it would be valuable to provide more details regarding the specific statistical tests used, assumptions made, and any post-hoc analyses conducted. Additionally, stating the software or programming language used for the analysis would be useful.

8.       What specific statistical tests were used to analyze the data?

9.       Were any post-hoc tests conducted to further analyze the significant differences?

Discussion:

The discussion section presents a comprehensive analysis and interpretation of the results obtained. The authors discuss the advantages of using C. reinhardtii as a host organism, such as its controllability, fast growth, and genetic characteristics. They highlight the successful application of gene loading and culture optimization strategies in increasing dammarenediol-II production. The effects of different culture conditions, including medium composition, light regime, and the addition of methyl jasmonate (MeJA), are discussed in relation to their impact on productivity. The challenges encountered in protein expression and potential future directions for improving protopanaxadiol production are addressed. The study's findings are compared with other microbial production systems for ginsenosides, providing a broader context for the research.
Overall, the discussion effectively communicates the key findings and their implications. However, there are a few suggestions for improvement:

·       Clarify the purpose of the study: The discussion could begin with a brief recap of the main objectives or hypotheses of the research to provide context for the subsequent points discussed.

·       Provide more specific details: In some instances, it would be helpful to provide specific details, such as the exact concentrations of MeJA used or the specific fusion constructs created, to enhance the clarity and precision of the discussion.

·       Include references for cited studies: The discussion references several studies to support the findings. It would be beneficial to include proper in-text citations or a reference list at the end to acknowledge the sources.

·       Proofread for clarity: There are a few sentences that could be rephrased or clarified to improve readability and ensure that the intended meaning is clear. For example, the sentence "We prolong the dodecane harvest time to 10 d, the accumulation of dammarenediol-II is highest extracted by dodecane when harvested at 7 d" could be revised for clarity.

1.       How do engineered microorganisms, including bacteria and yeasts, contribute to ginsenoside biosynthesis, and what are some examples of their

2.       How does the microalgal cellular environment contribute to the favorable production levels?

3.       What makes the C. reinhardtii strains engineered in this study a promising platform for further production of ginsenoside expression of plant terpene synthases (TPSs) compared to other hosts?

The quality of English language in the provided text is generally good. The sentences are well-structured, and the vocabulary is appropriate for a scientific paper. However, there are a few areas where improvements can be made.

Author Response

manuscript ID: ijms-2449083

Metabolic Promoting Photosynthetic Production of Dammarenediol-II in Chlamydomonas reinhardtii via Gene Loading and Culture Optimization

Reviewers' comments:

Reviewer 2

Comments and Suggestions for Authors

The paper describes a study that focuses on the production of dammarenediol-II, a triterpene compound found in Panax ginseng, using the microalgae Chlamydomonas reinhardtii as a production platform. The researchers introduced genes from Panax ginseng and Arabidopsis thaliana responsible for the synthesis of dammarane-type ginsenosides into Chlamydomonas reinhardtii to create a synthetic pathway for the production of these compounds. They employed strategies such as gene loading and culture optimization to enhance productivity.

The results showed that introducing multiple copies of the key enzyme gene, PgDDS, significantly increased the production of dammarenediol-II to approximately 0.2 mg/L. Furthermore, by optimizing the culture conditions using an opt2 medium supplemented with methyl jasmonate and a light:dark regimen, the titer of dammarenediol-II increased more than 13-fold to approximately 2.6 mg/L.

Overall, this study successfully engineered Chlamydomonas reinhardtii to produce dammarenediol-II, a promising candidate for pharmacologically active triterpenes found in Panax ginseng. The findings demonstrate the potential of microalgae as a platform for the production of ginsenosides. Further research and optimization could lead to the development of efficient and sustainable methods for producing these valuable bioactive compounds.

One potential improvement would be to provide more context regarding the significance of dammarenediol-II and dammarane-type ginsenosides in the pharmaceutical field. Additionally, it would be helpful to include any limitations or challenges encountered during the study. Overall, this paper provides a clear overview of the study's objectives, methods, and results, paving the way for further exploration in the field of ginsenoside production.

The review suggests a few improvements to enhance the paper's readability and flow before publication in the journal.
Introduction:

The introduction provides an overview of the importance of dammarenediol-II and ginsenosides as valuable compounds with various pharmaceutical and industrial applications. It mentions the challenges associated with traditional production methods and highlights the potential of microbial systems for efficient and sustainable production. The authors specifically focus on the microalga C. reinhardtii as a promising host organism due to its controllability, genetic characteristics, and well-established genetic manipulation techniques. The objectives of the study are clearly stated, emphasizing the use of gene loading and culture optimization to enhance dammarenediol-II production.

The introduction contains a few grammatical errors and areas that could be improved for clarity. Here are some suggested revisions:

  • "Plant natural terpenoids are seemingly important pharmaceuticals due to their anti-cancer (taxol and ginsenosides), anti-inflammatory (tanshinones), antimalarial (artemis- inin), and antioxidation (ginsenosides) properties [1–5]."

Revised: "Plant-derived natural terpenoids are considered important pharmaceuticals due to their anti-cancer properties (such as taxol and ginsenosides), anti-inflammatory effects (tanshinones), antimalarial activity (artemisinin), and antioxidative properties (including ginsenosides) [1–5]."

Response: Thank you for your valuable suggestions. We have revised the sentence.

  • "However, the high price of direct extraction from plants due to low amounts of the target molecule, is fluctuating in both quantity and quality, and negative environmental impact."

Revised: "However, the high cost of direct extraction from plants, coupled with the low quantities of the target molecule, fluctuating yields, and negative environmental impact..."

 Response: Thank you for your valuable suggestions. We have revised the sentence.

  • "And the complexity and chirality of natural terpenoids hampered their production by chemical synthesis [6]."

Revised: "Moreover, the complexity and chirality of natural terpenoids hinder their production through chemical synthesis [6]."

 Response: Thank you for your valuable suggestions. We have revised the sentence.

  • "With the advance of synthetic biology, isoprenoid engineering in microorganisms is becoming an alternative and attractive route for addressing the increasing demand for natural terpenoids [6]."

Revised: "The advancement of synthetic biology has made isoprenoid engineering in microorganisms an alternative and attractive route to meet the increasing demand for natural terpenoids [6]."

 Response: Thank you for your valuable suggestions. We have revised the sentence.

  1. Could you provide more insight into the specific challenges and limitations encountered in the traditional extraction methods of ginsenosides that prompted the exploration of alternative approaches?

Response: For instance, ginseng roots require approximately 6 years of growth until harvest, experiencing varying influences related to weather, soil, and the presence of pathogens. However, the total ginsenosides contents in 5- to 7-year-old P. ginseng roots are approximately 2% g/g dry weight, and some rare ginsenosides account for less than 0.01%, making it time-consuming and unsustainable by direct extraction. Moreover, due to the stereo-chemical complexity of ginsenosides, it is also challenging to synthesize by chemical method.

  1. How do the achieved dammarenediol-II titers in strain SV3 compare to previous studies or existing methods of ginsenoside production?

Response: A cell suspension culture of transgenic tobacco can produce 5.2 mg/L dammarenediol-II. Transgenic rice can produce 0.44 mg/g dw dammarenediol-II, 0.59 mg/g dw PPD, and 0.43 mg/g dw protopanaxatriol (PPT). The volumetric production of dammarenediol-II with squalene was up to 13.233 mg/L under the common shake flask fermentation conditions of transgenic Pichia pastoris. The highest dammarenediol-II titer was approximately 3.3 mg/L in transgenic algae when cultured with 1.5 mM MeJA in this study.

  1. Beyond ginsenosides, do you believe Chlamydomonas reinhardtii or similar microalgae platforms have potential for the production of other bioactive compounds, and if so, which ones and why?

Response: In industry, microalgae have been exploited to yield bioactive isoprenoids, such as β-carotene (from Dunaliella salina) and astaxanthin (from Haematococcus pluvialis). As a natural source of terpenoids, microalgae have a number of potential advantages over other biological resources. Thus, the endogenous bioactive isoprenoids in microalgae derive from CO2 and light. And the concentrations of natural carotenoids in microalgae exceed those found in other sources. For example, β-carotene in Dunaliella salina constitutes even 14% of dry weight. Microalgae also provide unique isoprenoids, which are not found in higher plants, such as fucoxanthin and astaxanthin. Previous researches have reported that the C. reinhardtii and Phaeodactylum tricornutum can be used to yield heterogenous terpenoids. For instance, patchoulol, 13R(+) manoyl oxide, (E)-α-bisabolene, labdane diterpenes, sclareol, geraniol, limonene, and betulin.

Materials and Methods:

The materials and methods section describes the experimental procedures and techniques used in the study. It includes information on the construction of gene cassettes and vectors, transformation of C. reinhardtii, gene loading strategy, culture conditions, and extraction methods for dammarenediol-II. The section provides sufficient detail to allow for replication of the experiments.

Some sentences are overly complex or contain multiple ideas, making them difficult to follow. Try breaking them into shorter, more concise sentences to improve readability.

Here is a list of grammatical problems in the text:

In the sentence "because of its capable of efficiently expressing transgenes," the word "capable" should be "capability."

Response: Thank you for your valuable suggestions. We have revised it.

In the sentence "UVM4 algal species, which originated from Prof. Dr. Ralph Bock, can grow well in freshwater environments," the word "which" should be "that."

Response: Thank you for your valuable suggestions. We have revised it.

In the sentence "We maintained this algal strain in continuous light (150 µmol m−2 s −1 ) at 25℃ environment," there should be a comma after "25℃" and before "environment."

Response: Thank you for your valuable suggestions. We have revised it.

In the sentence "The amino acid sequences for the dammarenediol synthase from P. ginseng (PgDDS, ACZ71036) [6], the PPD synthase, which is a CYP enzyme, from P. ginseng (PgCYP716A47, AEY75212) [7], and the P450 reductase 2 form A. thaliana (AtCPR, NP_194750) [6] were codon optimized," "form" should be "from."

Response: Thank you for your valuable suggestions. We have revised it.

In the sentence "We confirmed the vector sequences by sequencing (Sangon Biotech, Shanghai, China) following each cloning step," "following" should be "after" to indicate the sequence of events.

Response: Thank you for your valuable suggestions. We have revised it.

In the sentence "The TAP agar plates supplemented with 200 mg L−1 spectinomycin or/and 10 mg L−1 zeocin were used to select the positive transformants under a light intensity of 150 µmol photons m−2 s−1," "or/and" should be "and/or."

Response: Thank you for your valuable suggestions. We have revised it.

In the sentence "We further confirmed the transformants by using the genomic PCR and qRT-PCR methods," "by using" can be simplified to "using."

Response: Thank you for your valuable suggestions. We have revised it.

In the sentence "The dodecane fractions were collected by centrifugation at 8000 rcf for 6 min, then transferred to new sample tubes," "then transferred" should be "and then transferred."

Response: Thank you for your valuable suggestions. We have revised it.

In the sentence "After filtrating, the quantification of PPD and its intermediate products dammarenediol-

Response: After the filtration process, the quantification of PPD and its intermediate product, dammarenediol-II, in dodecane was conducted using gas chromatography-mass spectrometry (GC-MS).

In the sentence "The mass spectrometer was set to full-scan and sim mode," "sim mode" should be "SIM mode" or "selected ion monitoring mode."

Response: Thank you for your valuable suggestions. We have revised it.

In the sentence "All the fermentation processes were performed in triplicate," "All" can be removed to improve clarity.

Response: Thank you for your valuable suggestions. We have revised it.

Here are some questions related to this part of the Materials and Methods:

  1. Why was the C. reinhardtii UV-mutated 4 (UVM4) strain chosen for the experiments?

Response: Because of unsuitable codon usage in the open reading frame (ORF), the nuclear expression levels of nondomestic cDNA genes are disappointingly poor in wild-type C. reinhardtii. The C. reinhardtii UV-mutated 4 (UVM4, cell wall-deficient) strain can efficiently express transgenes and overcome the long-standing obstacle of the disappointing poor expression of transgenes in the algal nuclear genome.

  1. Where did the UVM4 algal species originate from, and what are its characteristics in terms of growth and environmental requirements?

Response: UVM4 algal species, that originated from Prof. Dr. Ralph Bock, can grow well in freshwater environments. We maintained this algal strain in continuous light (150 µmol m−2 s −1 ) at a 25℃, environment and cultured this strain by using a tris-acetate-phosphate (TAP) medium on agar plates or liquid culture.

  1. What medium was used to culture the UVM4 strain, and what were the conditions for maintaining it?

Response: We maintained this algal strain in continuous light (150 µmol m−2 s −1 ) at a 25℃, environment and cultured this strain by using a tris-acetate-phosphate (TAP) medium on agar plates or liquid culture.

  1. What were the specific modifications made to the PgDDS and PgCYP716A47 sequences, and why were they modified?

Response: In order to enable transgene expression of large codon-optimized constructs from the nuclear genome of C. reinhardtii, the PgDDS and PgCYP716A47 sequences were codon optimized for the nuclear codon bias of C. reinhardtii.

  1. What was the purpose of the two-phase extractive fermentation, and how was it performed?

Response: In order to capture the product dammarenediol-II from the cell wall-deficient C. reinhardtii UVM4 strain, the two-phase extractive fermentation was used. Added the 5 mL dodecane overlay into the 300 mL culture medium after the cells had reached the logarithmic phase.

  1. What optimization experiments were conducted to enhance dammarenediol-II production, and what were the specific conditions investigated?

It would be beneficial to mention the number of replicates performed for each experiment or assay. Additionally, providing information about the use of positive and negative controls would strengthen the experimental design.

Response: To investigate the influence of light regimen, we culture the strains under constant (24 h) light or 16:8 h L:D cycle conditions. To investigate the influence of the medium, we culture the strains in a TAP medium or opt2 medium (TAP medium + 1 mL L–1 glacial acetic acid). To investigate the influence of solvent overlay, we culture the strains in an opt2 medium supplemented with dodecane or perfluoro-2-butyl tetrahydrofuran. To investigate the influence of the different dodecane extraction times (4, 5, 6,7, 8, 9, and 10 d) on the dammarenediol-II production. The fermentation processes were performed in triplicate.

  1. Were there any specific controls or replicates used in the experiments, and if so, what were they?

While the statistical analysis is mentioned, it would be valuable to provide more details regarding the specific statistical tests used, assumptions made, and any post-hoc analyses conducted. Additionally, stating the software or programming language used for the analysis would be useful.

Response: The dodecane overlay of UVM4 was used as control, but no dammarenediol-II signal was detected, so we did not display it in the Figure. All fermentation processes and GC-MS determinations were performed in triplicate.

  1. What specific statistical tests were used to analyze the data?

Response: One-way ANOVA was used to analyze the significance of the differences.

  1. Were any post-hoc tests conducted to further analyze the significant differences?

Response: No.

Discussion:

The discussion section presents a comprehensive analysis and interpretation of the results obtained. The authors discuss the advantages of using C. reinhardtii as a host organism, such as its controllability, fast growth, and genetic characteristics. They highlight the successful application of gene loading and culture optimization strategies in increasing dammarenediol-II production. The effects of different culture conditions, including medium composition, light regime, and the addition of methyl jasmonate (MeJA), are discussed in relation to their impact on productivity. The challenges encountered in protein expression and potential future directions for improving protopanaxadiol production are addressed. The study's findings are compared with other microbial production systems for ginsenosides, providing a broader context for the research.
Overall, the discussion effectively communicates the key findings and their implications. However, there are a few suggestions for improvement:

  • Clarify the purpose of the study: The discussion could begin with a brief recap of the main objectives or hypotheses of the research to provide context for the subsequent points discussed.

Response: Thank you for your valuable suggestions. We have added the sentence “To develop an alternative approach for ginsenoside production, we established the biosynthesis pathways for dammarane-type ginsenosides in Chlamydomonas reinhardtii. Subsequently, we implemented multi-step strategies to enhance the production levels of dammarenediol-II.”.

  • Provide more specific details: In some instances, it would be helpful to provide specific details, such as the exact concentrations of MeJA used or the specific fusion constructs created, to enhance the clarity and precision of the discussion.

Response: Thank you for your valuable suggestions. The 1 mM, 1.5 mM, or 2 mM MeJA were used, and the production of dammarenediol-II increased 7-fold after 1.5 mM MeJA treatment. The SV3 strains mean transgenic algae harboring V1 (PPsad-PgDDS-TPsad and PFDX-PgCYP716A47-TFDX cassettes) and V2 (PPsad-PgDDS-TPsad, PFDX-PgCYP716A47-TFDX, and PFDX-AtCPR-TFDX cassettes) constructs.

  • Include references for cited studies: The discussion references several studies to support the findings. It would be beneficial to include proper in-text citations or a reference list at the end to acknowledge the sources.

Response: Thank you for your valuable suggestions. We have revised it.

  • Proofread for clarity: There are a few sentences that could be rephrased or clarified to improve readability and ensure that the intended meaning is clear. For example, the sentence "We prolong the dodecane harvest time to 10 d, the accumulation of dammarenediol-II is highest extracted by dodecane when harvested at 7 d" could be revised for clarity.

Response: Thank you for your valuable suggestions. We have rephased the sentences as “We collected dodecane fractions at various time points: 4, 5, 6, 7, 8, 9, and 10 days. The highest accumulation of dammarenediol-II was observed when extracted by dodecane after 7 days of growth”.

  1. How do engineered microorganisms, including bacteria and yeasts, contribute to ginsenoside biosynthesis, and what are some examples of their

Response: As shown in the last paragraph of the discussion. The most general strategies for yeasts to produce ginsenoside are through heterologous gene expression and enzyme engineering [9]. A cell suspension culture of transgenic tobacco can produce 5.2 mg/L dammarenediol-II [34]. Transgenic rice can produce 0.44 mg/g dw dammarenediol-II, 0.59 mg/g dw PPD, and 0.43 mg/g dw protopanaxatriol (PPT) [35]. The volumetric production of dammarenediol-II by feeding with squalene was up to 13.233 mg/L under the common shake flask fermentation conditions of transgenic Pichia pastoris [36].

  1. How does the microalgal cellular environment contribute to the favorable production levels?

Response: Light-driven photosynthetic microbes, such as Chlamydomonas reinhardtii can offer the potential for sustainable production processes [16,17]. Microalgal cellular environment is more favorable to the plant terpene synthases (TPSs) than bacterial, yeast, or cyano-bacterial hosts because they share evolutionary ancestry with land plants [17].

  1. What makes the C. reinhardtii strains engineered in this study a promising platform for further production of ginsenoside expression of plant terpene synthases (TPSs) compared to other hosts?

Response: Light-driven photosynthetic microbes, such as Chlamydomonas reinhardtii can offer the potential for sustainable production processes [16,17]. Microalgal cellular environment is more favorable to the plant terpene synthases (TPSs) than bacterial, yeast, or cyano-bacterial hosts because they share evolutionary ancestry with land plants [17].

Comments on the Quality of English Language

The quality of English language in the provided text is generally good. The sentences are well-structured, and the vocabulary is appropriate for a scientific paper. However, there are a few areas where improvements can be made.

Response: Thank you for your valuable suggestions. We have checked full text.

Reviewer 3 Report

Dear authors,

In this paper, the production of dammarenediol-II by Chlamydomonas has been addressed. However, the way the paper has been written and the experiments have been conducted has numerous weaknesses, inconsistencies, and unknowns that should be clearly addressed by the authors. One critical issue is the gene loading, or what they claim to be gene loading, as far as I know, they did not perform it.

Majors:

* When a foreign gene is inserted into the genome of Chlamydomonas, it occurs randomly and does not involve homologous recombination. Depending on the site of insertion, due to epigenetic phenomena, the same gene can have much higher expression in one strain compared to another. The authors do not actually verify the number of copies inserted in the genome of each strain. What I understand they do is a screening starting with 800-1000 transformants using qRT-PCR until they find some strains with higher expression of the introduced genes. The authors claim that it is due to the number of copies, but since they don't study it, they cannot prove it. Personally, I believe it is due to epigenetic phenomena.

L22: “Multiple copies of transgene expression cassettes were introduced into the genome” How was this multiple copy number quantified.

*L58-L64: It would be convenient for better understanding to include a figure of this metabolic pathway described here, showing the structures of the different intermediates. There is no reference throughout this paragraph, it must be an error.

*L103:  I do not understand why the transformants with PgCYP716A47 and/or AtCPR increased the amount of dammarenediol-II if dammarenediol-II is the substrate of PgCYP716A47 and/or AtCPR. Shouldn't a decrease be expected instead?

* In the screening of the 800 strains, they should have found many that expressed one construct and not the other. Why didn't they select and study them? Have they done so? Are there any differences between them in the expression of dammarenediol-II? I believe that those expressing only PgDDS and not PgCYP716A47 and/or AtCPR should have higher production of dammarenediol-II, or am I mistaken?

L113: “124 transformants were verified” how they were vverified? by qRT-PCR? please indicated it

L117: “have higher expression level of PgDDS and PgCYP716A47 among all of the SV1 strains, but they also have very low genes expression level” I don't understand this sentence, isn't it contradictory? It says one thing and the opposite in the same sentence.

*To indicate in the caption of Fig. 1 what "docecane overlay" means

*Fig 2D: SV3? `please check, a guest SV1

*The legend of Fig. 3 is inconsistent. Is it the "SV3" or the "SV1" strain? and which strain SV1-?? Why only one did you check more, please specified.

*L125: “No PPD signal was detected in any of the detected strains” poor english

*L143-L144: Why is it in bold? Moreover, grammatically, it is not correct. Please correct it.

*L196: “The predicted molecular masses of PgDDS (~90 kDa) and PgCYP716A47 (~56 kDa) were basically the same” I don't understand this statement. What do you mean by "the same"? There is a significant difference in molecular weight. Please explain.

*Fig 5D, the control of the load is missing.

*Table 2: Transformants which contained “with” V1 construct. Grammatically, it is not correct. I am just providing it as an example. The grammatical expressions and the English need a thorough revision.

*L90: “gene-loading to introducing multiple copies of a transgene expression cassette into the genome” I am sorry but, I have not found in the materials and methods, nor throughout the text, what the authors refer to with this technique "gene loading." This point should be clearly indicated. L172: “In order to improve the expression level, we introduced multiple copies of transgene expression cassettes introduced into the genome” yes, but how. Based on what do the authors claim to have achieved this? As far as I know, a technique like this has only a few times been described in Chlamydomonas to date. How did the authors manage to accomplish it? This point should be clearly clarified. It is because of this technique that the authors refer to as "gene loading" that one of the main reasons why this paper is interesting. In fact, the authors include it in the title. However, I'm afraid the authors do not demonstrate how they performed it or the results they obtained. How many extra copies of the genes do these supposedly obtained strains have using this technique? How was it quantified?

L172:” we introduced multiple copies of transgene” How many copies? How was this number verified?

*As you can see in Figure 5 D and E, in the case of the SV3-40 strain, the relative expression does not parallel the protein production. Is there any explanation for this discrepancy?

*L174: by sequential transformation? What does this mean?

*L180: “As we expected” I don't understand why they expect this. Can you please clarify it?

*L186: “But we did not examine the titer of PPD” Why? Explain please

*L223: after 3?? The data in the figure starts on day 4.

* It seems that MeJA 2 mM is somehow toxic, isn't it? Any idea why this is the case?

L261: “between SV3” which one?

* I don't understand what the metabolomic study contributes. It is not clearly indicated which strains were used and the chosen physiological conditions. And most importantly, why is this study not addressed in the discussion? What was the purpose of conducting it?

*L297: “which is the gene loading [28]” yes, but as far as I know not in Chlamydomonas.

*L300: “we also used the gene loading strategy “ I'm afraid this is not correct. The strategy used by the authors in Reference 29 has nothing to do with the one carried out in this paper.

L348: “We presumed that the codon-optimized PgCYP716A47 may not function in microalgae” I cannot find any sense in this statement. If you have detected protein expression in the western blot, why are you now saying that the use of codons is incorrect? Can you please clarify?

Minors:

*C. reinhardtii must always be in italics.

*To indicate in the caption of Fig. 4 what BLE stands for

*The legend of Figure 1 does not describe what it is PgCYP716A47

* Figure 6C, X-axis, indicate that it represents days.

Extensive editing of English language required

Author Response

manuscript ID: ijms-2449083

Metabolic Promoting Photosynthetic Production of Dammarenediol-II in Chlamydomonas reinhardtii via Gene Loading and Culture Optimization

Reviewers' comments:

Reviewer 3

Comments and Suggestions for Authors

Dear authors,

In this paper, the production of dammarenediol-II by Chlamydomonas has been addressed. However, the way the paper has been written and the experiments have been conducted has numerous weaknesses, inconsistencies, and unknowns that should be clearly addressed by the authors. One critical issue is the gene loading, or what they claim to be gene loading, as far as I know, they did not perform it.

Majors:

* When a foreign gene is inserted into the genome of Chlamydomonas, it occurs randomly and does not involve homologous recombination. Depending on the site of insertion, due to epigenetic phenomena, the same gene can have much higher expression in one strain compared to another. The authors do not actually verify the number of copies inserted in the genome of each strain. What I understand they do is a screening starting with 800-1000 transformants using qRT-PCR until they find some strains with higher expression of the introduced genes. The authors claim that it is due to the number of copies, but since they don't study it, they cannot prove it. Personally, I believe it is due to epigenetic phenomena.

L22: “Multiple copies of transgene expression cassettes were introduced into the genome” How was this multiple copy number quantified.

Response: Aw and Polizzi [1] and Lauersen, et al. [2] reported that generating a strain carrying multiple cognate genes is to use different selection markers for sequential integration. We have a sequential integrated V1 vector (which harbors PPsad-PgDDS-TPsad and PFDX-PgCYP716A47-TFDX cassettes) and V2 vector (which harboring PPsad-PgDDS-TPsad, PFDX-PgCYP716A47-TFDX, and PFDX-AtCPR-TFDX cassettes) into C. reinhardtii, so we thought introduced multiple copies of the transgene.

*L58-L64: It would be convenient for better understanding to include a figure of this metabolic pathway described here, showing the structures of the different intermediates. There is no reference throughout this paragraph, it must be an error.

Response: Thank you for pointing out our neglect. We have added the reference in this paragraph and described the metabolic pathway in Figure 1A.

*L103:  I do not understand why the transformants with PgCYP716A47 and/or AtCPR increased the amount of dammarenediol-II if dammarenediol-II is the substrate of PgCYP716A47 and/or AtCPR. Shouldn't a decrease be expected instead?

Response: First of all, our purpose is to construct the dammarane-type ginsenosides synthetic pathway in C. reinhardtii, not only produce dammarenediol-II. So, we constructed the transformants with PgDDS, PgCYP716A47, and/or AtCPR. As Dai, et al. [3] reported that engineered yeast strains can accumulate a large amount of dammarenediol-II, but could not increase protopanaxadiol production by increasing copy numbers of PPDS (also known as CYP716A47) and CPR. Thus, the transformants with PgCYP716A47 and/or AtCPR can increase the amount of dammarenediol-II.

* In the screening of the 800 strains, they should have found many that expressed one construct and not the other. Why didn't they select and study them? Have they done so? Are there any differences between them in the expression of dammarenediol-II? I believe that those expressing only PgDDS and not PgCYP716A47 and/or AtCPR should have higher production of dammarenediol-II, or am I mistaken?

Response: Thank you for your valuable suggestions. It is a good question!

Due to the random integration of transgenes in chlamydomonas, there are great differences in the expression level among transformants. We detected a large number of transformants to obtain engineering algae with high expression of PgDDS, PgCYP716A47, and/or AtCPR. In the Results 2.1 section, we have investigated the transformants which only expressed V1 or V2 vector, the titer of dammarenediol-II approximately 30 µg/L of SV1 strains (only expressed V1 vector) or 70 µg/L of SV2 strains (only expressed V2 vector). Our purpose is to construct the dammarane-type ginsenosides synthetic pathway in C. reinhardtii, so we neglected the transformants which only expressed PgDDS. Yes, the transformants which expressed only PgDDS and not PgCYP716A47 and/or AtCPR should have higher production of dammarenediol-II.

L113: “124 transformants were verified” how they were verified? by qRT-PCR? please indicated it

Response: The 124 transformants were verified by PCR.

L117: “have higher expression level of PgDDS and PgCYP716A47 among all of the SV1 strains, but they also have very low genes expression level” I don't understand this sentence, isn't it contradictory? It says one thing and the opposite in the same sentence.

Response: Thank you for your valuable suggestions. We have revised the sentence as “The strains SV1-12, SV1-25, and SV1-33 (transformants containing PPsad-PgDDS-TPsad and PFDX-PgCYP716A47-TFDX cassettes) exhibited higher expression levels of PgDDS and PgCYP716A47 compared with other SV1 strains. However, the gene expression levels of PgDDS and PgCYP716A47 were still considerably low when compared to the endogenous actin gene (Figure 2B).”

*To indicate in the caption of Fig. 1 what "docecane overlay" means

Response: The dodecane overlay is the extraction solvent, which can capture the  target products.

*Fig 2D: SV3? `please check, a guest SV1

Response: Thank you for pointing out our neglect. We have revised it.

*The legend of Fig. 3 is inconsistent. Is it the "SV3" or the "SV1" strain? and which strain SV1-?? Why only one did you check more, please specified.

Response: Thank you for your valuable suggestions. It is SV1-12 strain. We have checked all of the selected SV1 strains, but just show one as a representative.

*L125: “No PPD signal was detected in any of the detected strains” poor English

Response: Thank you for your valuable suggestions. We have revised the sentence as “No PPD signal was detected among all of the detected strains”.

*L143-L144: Why is it in bold? Moreover, grammatically, it is not correct. Please correct it.

Response: Thank you for your valuable suggestions. We have revised it.

*L196: “The predicted molecular masses of PgDDS (~90 kDa) and PgCYP716A47 (~56 kDa) were basically the same” I don't understand this statement. What do you mean by "the same"? There is a significant difference in molecular weight. Please explain.

Response: Thank you for your valuable suggestions. We have revised the sentence as “Full-length expression of each protein (PgDDS (~90 kDa) and PgCYP716A47 (~56 kDa)) could be detected.”.

*Fig 5D, the control of the load is missing.

Response: The JUV means UVM4, we have renamed the JUV as UVM4 in new Figure 5, thank you for pointing out our neglect.

*Table 2: Transformants which contained “with” V1 construct. Grammatically, it is not correct. I am just providing it as an example. The grammatical expressions and the English need a thorough revision.

Response: Thank you for your valuable suggestions. We have revised it.

*L90: “gene-loading to introducing multiple copies of a transgene expression cassette into the genome” I am sorry but, I have not found in the materials and methods, nor throughout the text, what the authors refer to with this technique "gene loading." This point should be clearly indicated. L172: “In order to improve the expression level, we introduced multiple copies of transgene expression cassettes introduced into the genome” yes, but how. Based on what do the authors claim to have achieved this? As far as I know, a technique like this has only a few times been described in Chlamydomonas to date. How did the authors manage to accomplish it? This point should be clearly clarified. It is because of this technique that the authors refer to as "gene loading" that one of the main reasons why this paper is interesting. In fact, the authors include it in the title. However, I'm afraid the authors do not demonstrate how they performed it or the results they obtained. How many extra copies of the genes do these supposedly obtained strains have using this technique? How was it quantified?

Response: Aw and Polizzi [1] and Lauersen, et al. [2] reported, gene-loading is involving either high-copy plasmids or multiple copies of a transgene expression cassette introduced into the genome. In this study, we introduced V1 (which harbors PPsad-PgDDS-TPsad and PFDX-PgCYP716A47-TFDX cassettes) vector into C. reinhardtii to obtain SV1 strain, and then introduced the V2 vector (which harbors PPsad-PgDDS-TPsad, PFDX-PgCYP716A47-TFDX, and PFDX-AtCPR-TFDX cassettes) into the SV1 strain, by double transformation and selection mediated by separate antibiotic resistance markers. So, we thought this strategy is gene loading.

L172:” we introduced multiple copies of transgene” How many copies? How was this number verified?

Response: As Aw and Polizzi [1] and Lauersen, et al. [2] reported that generating a strain carrying multiple cognate genes is to use different selection markers for sequential integration. We have a sequential integrated V1 vector (which harbors PPsad-PgDDS-TPsad and PFDX-PgCYP716A47-TFDX cassettes) and V2 vector (which harbors PPsad-PgDDS-TPsad, PFDX-PgCYP716A47-TFDX, and PFDX-AtCPR-TFDX cassettes) into C. reinhardtii, so we thought introduced multiple copies of the transgene.

*As you can see in Figure 5 D and E, in the case of the SV3-40 strain, the relative expression does not parallel the protein production. Is there any explanation for this discrepancy?

Response:Figure 5E shows the proteins of PgDDS and PgCYP716A47, Figure 5D shows the expression level of AtCPR. There is no relationship between them Figure 5E shows that the expression level of PgDDS is similar with the protein production, except SV3-15; the PgCYP716A47 expression level of SV3-141 is the highest of SV3 strains, and only SV3-141 protein production was detected.

*L174: by sequential transformation? What does this mean?

Response: We went through two rounds of transformation. First, introducing the V1 vector (which harborss PPsad-PgDDS-TPsad and PFDX-PgCYP716A47-TFDX cassettes) into C. reinhardtii to obtain SV1 strain; second, introducing V2 (which harboring PPsad-PgDDS-TPsad, PFDX-PgCYP716A47-TFDX, and PFDX-AtCPR-TFDX cassettes) vector into the SV1 strain.

*L180: “As we expected” I don't understand why they expect this. Can you please clarify it?

Response: As Aw and Polizzi [1] and Lauersen, et al. [2] reported that generating a strain carrying multiple cognate genes is to use different selection markers for sequential integration. We have a sequential integrated V1 vector (which harbors PPsad-PgDDS-TPsad and PFDX-PgCYP716A47-TFDX cassettes) and V2 vector (which harborss PPsad-PgDDS-TPsad, PFDX-PgCYP716A47-TFDX, and PFDX-AtCPR-TFDX cassettes) into C. reinhardtii, so we expected the expression level of PgDDS and PgCYP716A47 to be higher in SV3 strains.

*L186: “But we did not examine the titer of PPD” Why? Explain please

Response: Thank you for your valuable suggestions. We have examined the PPD using the GC-MS method but did not detect any signal of the PPD. We presumed that the large size of constructs (V1 and V2) may affect the folding and natural conformation of the multidomain protein. And the heterologous expression of CYP716A47 in microorganisms will result in low coupling between CPR, leading to reduced growth and terpenoid synthesis.

*L223: after 3?? The data in the figure starts on day 4.

Response: Thank you for your valuable suggestions. We have revised it.

* It seems that MeJA 2 mM is somehow toxic, isn't it? Any idea why this is the case?

Response: Yes, the 2 mM MeJA is toxic, MeJA treatment triggered oxidative stress, arrested growth, and altered the photosynthetic activity of the cells.

L261: “between SV3” which one?

* I don't understand what the metabolomic study contributes. It is not clearly indicated which strains were used and the chosen physiological conditions. And most importantly, why is this study not addressed in the discussion? What was the purpose of conducting it?

Response: Thank you for your valuable suggestions. The SV3 means SV3-141 strain. The metabolomic study shows that except for target products there are many other metabolites that have changed after transformation in C. reinhardtii. As shown in this study, some of the changed metabolites have medicinal properties but need further study to identify.

*L297: “which is the gene loading [28]” yes, but as far as I know not in Chlamydomonas.

Response: As shown in Reference 29, they used a gene-loading strategy to produce patchoulol in C. reinhardtii.

*L300: “we also used the gene loading strategy “ I'm afraid this is not correct. The strategy used by the authors in Reference 29 has nothing to do with the one carried out in this paper.

Response: Thank you for your valuable suggestions. As shown in Reference 29, “One strategy in overexpression of target proteins is gene-loading, involving either high copy plasmids or multiple copies of a transgene expression cassette introduced into the genome”. In this study, we sequentially transformed the V1 vector (which harbors PPsad-PgDDS-TPsad and PFDX-PgCYP716A47-TFDX cassettes) and V2 vector (which harbors PPsad-PgDDS-TPsad, PFDX-PgCYP716A47-TFDX, and PFDX-AtCPR-TFDX cassettes) into C. reinhardtii. So, we have generated multiple copies of a transgene expression cassette introduced into the genome.

L348: “We presumed that the codon-optimized PgCYP716A47 may not function in microalgae” I cannot find any sense in this statement. If you have detected protein expression in the western blot, why are you now saying that the use of codons is incorrect? Can you please clarify?

Response: Thank you for your valuable suggestions. We have deleted this sentence.

Minors:

*C. reinhardtii must always be in italics.

Response: Thank you for your valuable suggestions. We have checked the full text.

*To indicate in the caption of Fig. 4 what BLE stands for

Response: Thank you for your valuable suggestions. We have revised the legend of Figure 4.

*The legend of Figure 1 does not describe what it is PgCYP716A47

Response: Thank you for your valuable suggestions. We have revised the legend of Figure 1.

* Figure 6C, X-axis, indicate that it represents days.

Response: Thank you for your valuable suggestions. We have revised Figure 6C.

Comments on the Quality of English Language

Extensive editing of English language required

Response: Thank you for your valuable suggestions. We have checked full text.

  1. Aw, R.; Polizzi, K.M. Can too many copies spoil the broth? Microbial Cell Factories 2013, 12, 1-9.
  2. Lauersen, K.J.; T., B.; J., W.; R., W.; H., M.J.; W., H.; T., H.; O., K. Efficient phototrophic production of a high-value sesquiterpenoid from the eukaryotic microalga Chlamydomonas reinhardtii. Metab. Eng. 2016, 38, 331-343, doi:10.1016/j.ymben.2016.07.013.
  3. Dai, Z.; Liu, Y.; Zhang, X.; Shi, M.; Wang, B.; Wang, D.; Huang, L.; Zhang, X. Metabolic engineering of Saccharomyces cerevisiae for production of ginsenosides. Metab Eng 2013, 20, 146-156, doi:10.1016/j.ymben.2013.10.004.

Round 2

Reviewer 2 Report

The Authors addressed all comments raised by reviewers and the manuscript has been revised  accordingly . The revisions have resulted in the significant improvement of the manuscript.

Reviewer 3 Report

Dear authors,

I believe the authors have done a good job in addressing all my doubts correctly, and I accept the paper in its current version.